# A Novel Channel Preparation Scheme to Optimize Program Disturbance in Three-Dimensional NAND Flash Memory

**DOI:** 10.3390/mi15020223

**Published:** 2024-01-31

**Authors:** Kaikai You, Lei Jin, Jianquan Jia, Zongliang Huo

**Affiliations:** 1Institute of Microelectronics, Chinese Academy of Sciences, Beijing 100029, China; youkaikai20@mails.ucas.ac.cn (K.Y.); huozongliang@ime.ac.cn (Z.H.); 2University of Chinese Academy of Sciences, Beijing 100049, China

**Keywords:** 3D NAND flash, GIDL pre-charge, program disturbance

## Abstract

The program disturbance characteristics of three-dimensional (3D) vertical NAND flash cell array architecture pose a critical reliability challenge due to the lower unselected word line (WL) pass bias (Vpass) window. In other words, the key contradiction of program disturbance is that the operational Vpass during the program’s performance cannot be too high or too low. For instance, the 3D NAND program’s operation string needs a lower Vpass bias to suppress unselected WL Vpass bias-induced Fowler–Nordheim tunneling (FN tunneling), but for the inhibited string, the unselected WL needs a higher Vpass bias to suppress selected WL program bias (Vpgm)-induced FN tunneling. In this paper, a systematical insight into the relationship between the channel potential and channel electron density is given. Based on this intensive investigation, we studied a novel channel preparation scheme using “Gate-induced drain leakage (GIDL) pre-charge”. Our methodology does not require the introduction of any new structures in 3D NAND, or changes in the operational Vpass bias. Instead, the potential on the unselected channel is enhanced by exploiting the holes generated by the GIDL operation effectively, leading to significantly suppressed program disturbance and a larger pass disturb window. To validate the effectiveness of the “GIDL pre-charge” method, TCAD simulation and real silicon data are used.

## 1. Introduction

Three-dimensional (3D) stackable NAND flash memory was invented to uphold the trends of increasing bit density and decreasing bit cost for each generation of NAND flash product [1,2,3]. To further achieve a minimal cell footprint and die size, the 3D NAND architecture with CMOS under an array replaced the CMOS below the NAND array. Typically, after a CMOS is fabricated, the 3D NAND array strings land on the n+ region, and the erase operation is accomplished by applying “GIDL-assisted body biasing” through tremendous electron–hole pair generation [4]. Caillat et al. proposed an optimization method for GIDL erase with dual-side (bottom selected gate side and top selected gate side) GIDL injection for a larger number of WLs. Tao et al. have investigated the correlation between GIDL erasure and the number of layers using TCAD simulation [4,5,6,7] and proved the feasibility of GIDL erasure for hundreds of layers. However, with the number of WL layers increasing to hundreds, the program strings suffered from pass disturbance, and the inhibited strings suffered from program disturbance, becoming a non-negligible issue during the program’s operation.

During a program’s operation, an unselected WL is the biased Vpass, and the selected WL is the biased Vpgm. Therefore, the program string suffers Vpass disturbance, and inhibited string suffers program disturbance [8,9]. As the number of WL layers increases, the Vpass disturbance becomes stronger due to more frequent Vpass stress. In this paper, we use TCAD simulation to study the mechanism of the “GIDL pre-charge” scheme. In the proposed scheme, GIDL generates the electron–hole pairs, and the generated holes facilitate the elimination of residential electrons in the poly channel hole. In this way, both pass disturbance and program disturbance are alleviated simultaneously since fewer electrons in the inhibited string elevate the channel potential.

## 2. Materials and Methods

Before introducing the conventional channel preparation scheme, an n+ region 3D NAND structure is shown in Figure 1a, which consists of the word line (WL), top/bottom selected transistor (TSG/BSG), source line (SL), and bit line (BL). For the efficiency of generating GIDL, the channel under BSG/TSG was also covered by highly doped phosphorus (the phosphorus doping concentration is about 1 × 10^19^ cm^−3^) and directly connected to the bottom n+ region (the phosphorus doping concentration is about 1 × 10^18^ cm^−3^). Figure 1b shows the corresponding TCAD simulation structure. In order to simplify the simulation, we just chose the 64 WL layer; Table 1 shows the detailed device structure parameters of the simulation. For simplification, the three TSG/BSG layers and two top/bottom dummy (DMY) layers are not depicted in detail in Figure 1, and the simulation setting is almost the same as in our previous work [10,11]. The electric potential and electron density along the channel of the conventional scheme and novel scheme were simulated using the Synopsys Sentaurus Sdevice simulator. The models used in TCAD device simulation are as follows: the Shockly–Read–Hall (SRH) model, non-local tunneling (NLT) model, Poole–Frenkel model, and Drift-Diffusion model [5]. These models can effectively reflect the physical characteristics and have proven useful for explaining many phenomena of 3D NAND flash.

In this work, silicon experiments were conducted on packaged 3D NAND test chips with n+ region architecture, for which the schematic of one block is shown in Figure 2a. And the test flow of array-level program disturbance is described in Figure 2b. First, the memory cells from WL0 to WLn − 1 of all the strings were programmed with random triple-level-cell (TLC) patterns after block erasure, and then the string0 WLn L0 Vt distributions were collected and read. Second, the memory cells in WLn of all the strings were programmed with random TLC patterns, and then the string0 WLn was collected and read. The array-level program distributions between the first read and the second read (denoted as ∆Vt) are explained in Figure 2c. A larger ∆Vt would induce a higher fail bit count during the read operation [12]; hence suppressing ∆Vt (program disturbance) can significantly ameliorate the reliability of 3D NAND flash. 

## 3. Experiments and Simulations

In the n+ region 3D NAND flash, the GIDL erasure operation was performed by applying high positive voltages to the BL/SL, while the TSG/BSG remains at a relatively low voltage. Then, B2B tunneling happened in the n+ region of TSG/BSG, which resulted in hole–electron pair generation. Meanwhile, the holes were transported into the channel in the electrical field direction, leading to the elevation of the channel potential. Afterward, the injected hole was transported into the gate stack of each cell, and the corresponding Vt of the cells shifted down. Based on this basic erasure operation, this paper a proposes GIDL pre-charge operation, for which the biasing is similar to that of the GIDL erasure operation. 

To differentiate between the conventional and proposed pre-charge schemes during the program’s operation, an inhibited string operation timing diagram is shown in Figure 3. Figure 3a shows a conventional pre-charge scheme. During the pre-charge phase, the SL and the BSG of the NAND strings are biased at Vsl (~2 V) and Vbsg (~4 V), respectively. And the other WLs are all biased at 0 V (including the BL and TSG). During program phase, the BL, SL, TSG, and BSG are biased at 0 V, and the other WLs are biased at Vpass (~9 V), and the selected WL (Sel WL) is biased at Vpgm (~20 V). For the program string, the TSG bias is about 4 V to turn on the TSG, and the channel potential is ~0 V, and the unselected WL suffers FN tunneling due to the Vpass bias, which leads to Vpass disturbance. Figure 3b presents a detailed waveform diagram of the proposed “GIDL pre-charge” program scheme. The SL and BL are biased to Vgidl (~5 V), while the TSG and BSG are turned off. The overall biasing condition is similar to GIDL erasure. These special biases lead to a similar GIDL effect (hole–electron pair generation) during the pre-charge phase. And during the program phase, the bias is the same as in Figure 3a.

TCAD simulations were performed with an n+ region 3D NAND structure to study the program disturbance of the WLn, following the experimental conditions described above: This program sequence ran from the TSG side to the BSG side. All the WLs below the WLn (Sel. WLn) were programmed to a random pattern, and the WLs above WLn were erased in a pattern because of the block erase operation (Vt < −2 V). Figure 4a illustrates the simulation results of channel global boosting potential with the conventional pre-charge scheme. During the pre-charge phase (T0–T3), the channel potential is elevated by the SL bias, which increases the WLn self-boosting channel potential. During the program’s operation (T4–T5), the self-boosting channel potential is decreased due to the depletion region disappearing [13,14]. In Figure 4b, the channel potential of the proposed scheme during the pre-charge phase (T0–T3) is different from the conventional pre-charge scheme. As shown in Figure 4a, the conventional pre-charge scheme only boosts the erased pattern channel potential during T0 and T3; there is a limitation of Sel. WLn self-boosting from T4 to T5. But in Figure 4b, the programmed random pattern cell and erased pattern cell channel potentials all boost at T1; it is clear that a high channel potential appeared from the TSG and BSG sides, and then all the cell channel potentials stabilized at T3. The proposed pre-charge scheme channel potential is about 2 V higher than the conventional pre-charge scheme in Figure 4a (T3 line). This higher channel potential remained during the following program phase (T4). The higher the channel potential is during the program, the less the program disturb will affect the threshold voltage of the selected WLs cells.

Furthermore, the TCAD simulation of the proposed scheme was performed to check the underlying cause of improvement program disturbance for the inhibited string, as given in Figure 5. And Figure 6 shows the TCAD simulation structure of band-to-band generation at T1 with two different pre-charge schemes. Figure 5a shows that identified that channel potential is affected by channel electron density. At the beginning of pre-charge phase, the channel has a large electron density because of the down-coupling effect [15,16], the residual electrons in the channel accumulate around erase cell, and these electrons decrease the channel potential at the beginning of the program phase (T4). During the conventional pre-charge phase, Vbsg turns on the BSG, and the Vsl elevated above the WLn + 1 channel potential. At T3, it is obvious that the channel electron density decreased throughout the pre-charge phase because of the higher channel potential. Figure 5b shows the channel electron density during the proposed pre-charge phase. After the BL/SL bias ramp to Vgidl, GIDL generated electron–hole pairs in the channel near TSG/BSG, and the hole drifted and diffused into the channel, which lead to a channel electron density decrease and hole density increase. Therefore, channel potential was elevated by the GIDL-generated hole. Compared with Figure 4a,b (T3 line), the proposed pre-charge scheme has a higher channel potential and a lower channel electron density than the conventional pre-charge scheme. Figure 5b,c shows the TCAD simulation structure inside of the dashed square box of Figure 5a,b at T0 and T3, respectively. It is clear that the conventional pre-charge scheme erase pattern channel electron flowed out in the direction of SL, but the program random pattern channel electron was almost unchanged. On the contrary, the proposed pre-charge scheme program pattern and erased pattern channel electrons flowed out through the BL/SL in two directions. Furthermore, Figure 6 illustrates the SL-side B2B generation at T1, and the proposed pre-charge scheme B2B generation is about 1 × 10^25^ cm^−3^/s, which subsequently decreased the number of channel electrons and elevated the channel potential. 

Figure 7a illustrates the array-level program disturbance of WLn; the experimental data show that the WLn experienced stronger program disturbance, which lead to severe array reliability challenges. To suppress the program disturbance and Vpass disturbance, a novel program scheme is proposed for n+ region 3D NAND flash. Correspondingly, the array level program disturbance suppressing proposed “GIDL pre-charge” is shown Figure 7b. There are 80 programming pulses between the first reading and the second reading. Compared with Figure 7a, the first read for the collected Vt distribution almost coincides with second read, which means the proposed program scheme could eliminate program disturbance during the program’s operation.

To summarize, TCAD simulation is used for the analysis of the changes in channel self-boosting potential and the channel electron/hole density during T0–T5. The proposed GIDL pre-charge program scheme can effectively eliminate the channel residual electrons and improve the channel self-boosting potential. Correspondingly, the program disturbance of the WLs is improved.

## 4. Conclusions

To mitigate the worse pass disturbance due to the increased number of layers, in this work, we propose a new pre-charge scheme based on the GIDL effect. Compared with the conventional scheme, the proposed scheme can better reduce the number of channel electrons, raise the pre-charging potential, and further improve the program disturbance and pass disturbance. To validate the efficiency of the proposed pre-charge scheme, TCAD simulation indicates a higher boosting channel potential in the inhibited string. Furthermore, our silicon experiment indicates that the Vt shift due to the program disturbance is significantly suppressed by exploiting the proposed pre-charge scheme. In this paper, a deep insight into the fundamental mechanism behind the proposed scheme and the conventional one is also included. The conventional pre-charge scheme is limited when suppressing the program disturbance because of the residual electrons in the poly channel during pre-charging. More specifically, the conventional pre-charge scheme can only reduce the channel electrons in the erasure pattern and lift the channel potential in the erasure pattern. For the program pattern, the electrons cannot directly flow out because the program pattern is turned off. By contrast, the proposed GIDL pre-charge scheme can either eliminate the residual electrons in the erasure pattern or the programmed pattern. Consequently, both the channel potential of erasure pattern and the programmed pattern are elevated since the generated hole by GIDL compounds the residual electrons. The increased channel potential of the program pattern after the pre-charge phase results in a higher channel self-boosting potential during the program’s operation, and thereby improves the program disturbance of the WLs.

## Figures and Tables

**Figure 1 micromachines-15-00223-f001:**
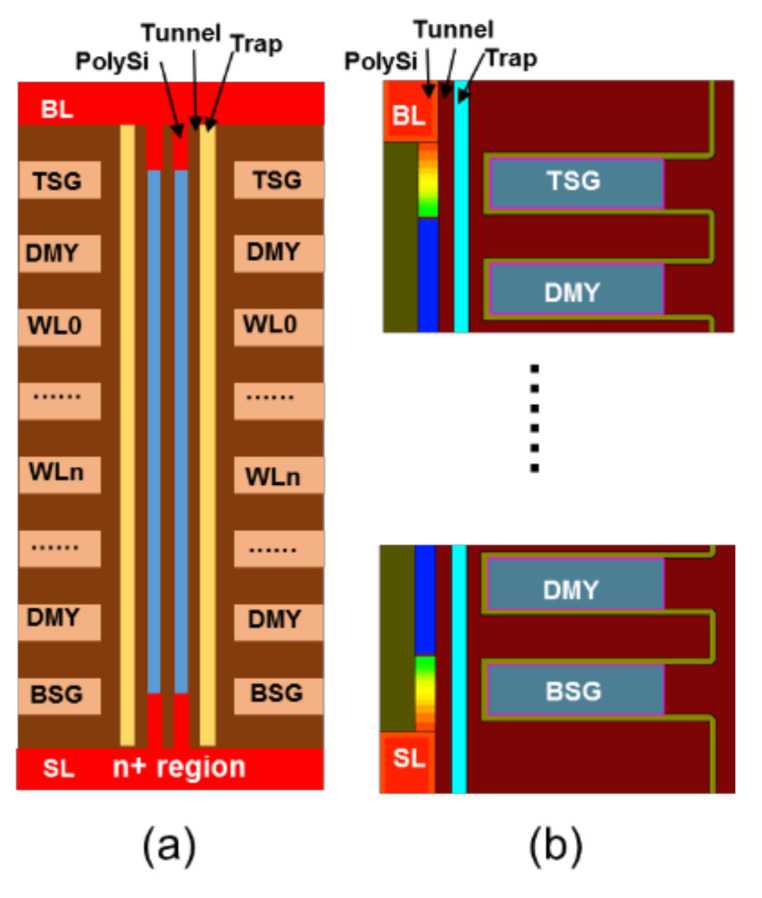
(**a**) Schematic cross-section of 3D NAND strings under n+ region. (**b**) TCAD simulation structure.

**Figure 2 micromachines-15-00223-f002:**
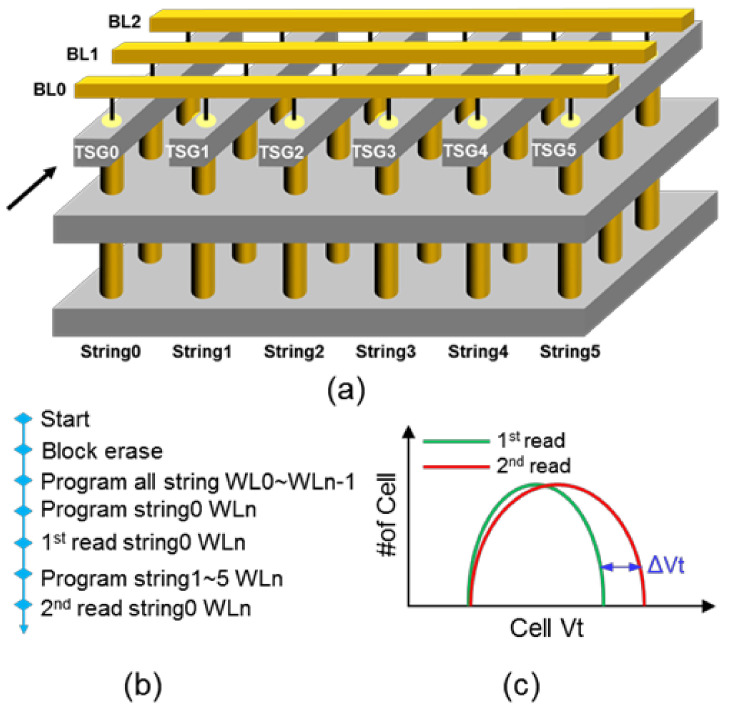
(**a**) Schematic illustration of one block in 3D NAND flash. (**b**) The testing approach and (**c**) characterization of array-level program disturbance.

**Figure 3 micromachines-15-00223-f003:**
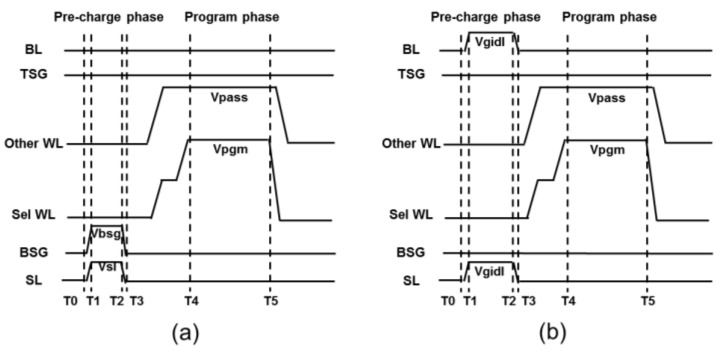
(**a**) The conventional signal timing diagrams of the program’s operation. (**b**) The proposed signal timing diagrams of the program’s operation.

**Figure 4 micromachines-15-00223-f004:**
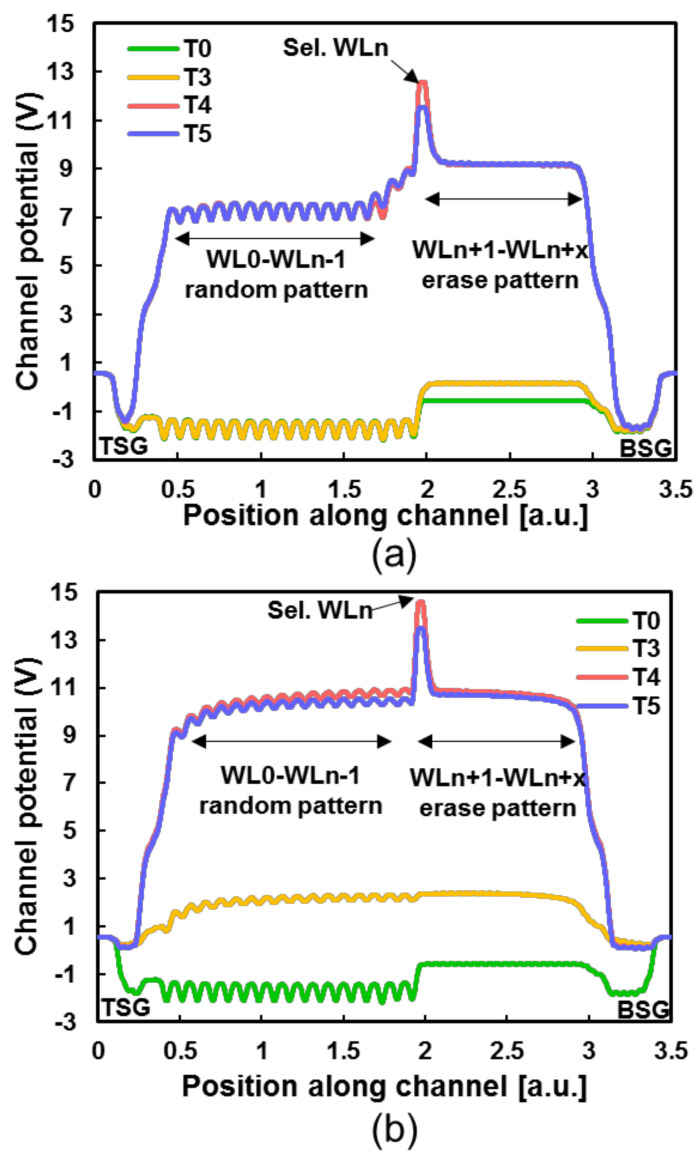
(**a**) TCAD simulation results of conventional pre-charge scheme channel boosting potential. (**b**) TCAD simulation results of proposed pre-charge scheme channel boosting potential.

**Figure 5 micromachines-15-00223-f005:**
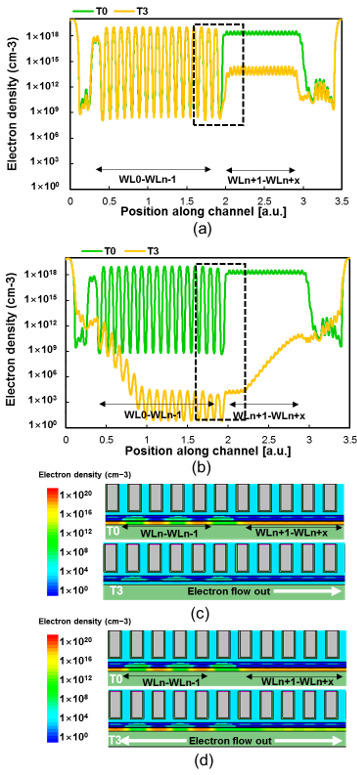
TCAD simulation results of conventional (**a**) and proposed (**b**) pre-charge scheme channel electron densities. TCAD simulation structure of conventional (**c**) and proposed (**d**) pre-charge scheme channel electron densities at T0 and T3, respectively.

**Figure 6 micromachines-15-00223-f006:**
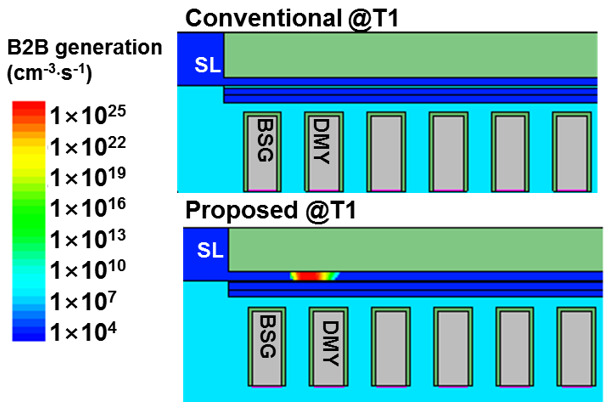
TCAD simulation structure of conventional and proposed scheme B2B generation @T1.

**Figure 7 micromachines-15-00223-f007:**
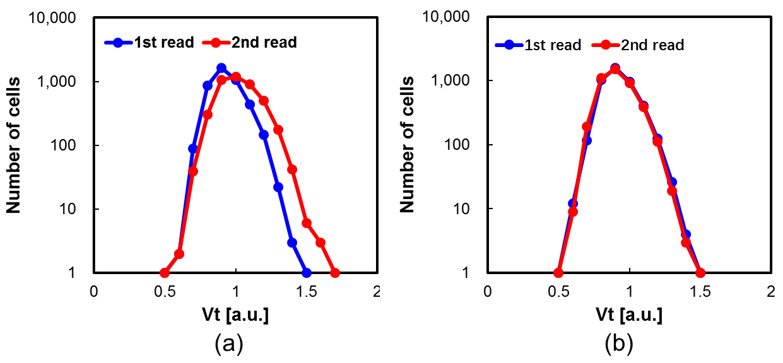
(**a**) Array level of Vt distribution for first read and second read. (**b**) Array level of Vt distribution for first read and second read with proposed program scheme.

**Table 1 micromachines-15-00223-t001:** Main parameters used in the simulation.

Parameter	Value
Gate length	33 nm
Gate spacing	21 nm
O/N/O	4/8/4
Channel hole diameter	80 nm
Poly-Si channel thickness	9 nm
WL layer	64
TSG layer	3
BSG layer	3
DMY layer	4

## Data Availability

Data is contained within the article.

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
