# Peer review of "A Novel Channel Preparation Scheme to Optimize Program Disturbance in Three-Dimensional NAND Flash Memory"

_micromachines, 2024, doi:10.3390/mi15020223_

Round 1

Reviewer 1 Report

Comments and Suggestions for Authors

1.In fig5 (c) and (d) “WLn-WLn-1” might be a typing-error. These figures are too small and are not easy to see the detail.

2 The program disturbance is a cumulative effect, the program operation numbers between 1st read ang 2nd read showed be indicated in fig7.

Author Response

Reviewer: 1.In fig5 (c) and (d) “WLn-WLn-1” might be a typing-error. These figures are too small and are not easy to see the detail.

Our response: 
Thank you for your reminding. I have correct my typing-error and enlarge the figures try to let them easy to see the detail. 

 Reviewer: 2 The program disturbance is a cumulative effect, the program operation numbers between 1st read and 2nd read showed be indicated in fig7.
Our response: 
  Thank you for your comment. There are 80 programming pulses between the 1st read and the 2nd read. I have added it in my text.

Reviewer 2 Report

Comments and Suggestions for Authors

Boosting channel potential in the inhibited string by GIDL appears applicable to prevent PGM disturbance. However, the pre-charge scheme inevitably requires device modification, which differs from the authors' understanding. In contrast to GIDL erasing, frequent programming using GIDL is critical in terms of stress and cell reliability, such as hot carriers. Hence, it would be better if the authors provided recommendations for device modification to address this concern.

Author Response

Reviewer: Boosting channel potential in the inhibited string by GIDL appears applicable to prevent PGM disturbance. However, the pre-charge scheme inevitably requires device modification, which differs from the authors' understanding. In contrast to GIDL erasing, frequent programming using GIDL is critical in terms of stress and cell reliability, such as hot carriers. Hence, it would be better if the authors provided recommendations for device modification to address this concern.

Our response:
Thank you for your comment. Yes, the pre-charge scheme will be induced TSG/BSG Vt shift down because of the hot carriers (GIDL induced hot hole injected into TSG/BSG). In order to decrease this phenomenon, we would use this scheme in the last PGM pulse and the Vgidl is about 4~5V, and in the former program pulse, we use ACS pre-charge to suppress the program disturbance.